# An Optimal Subspace Deconvolution Algorithm for Robust and High-Resolution Beamforming

**DOI:** 10.3390/s22062327

**Published:** 2022-03-17

**Authors:** Xiruo Su, Qiuyan Miao, Xinglin Sun, Haoran Ren, Lingyun Ye, Kaichen Song

**Affiliations:** 1College of Biomedical Engineering & Instrument Science, Zhejiang University, Hangzhou 310058, China; xiruo_su@zju.edu.cn (X.S.); 11715040@zju.edu.cn (Q.M.); 2School of Earth Sciences, Zhejiang University, Hangzhou 310058, China; rhr@zju.edu.cn; 3School of Aeronautics and Astronautics, Zhejiang University, Hangzhou 310058, China; kcsong@zju.edu.cn

**Keywords:** Direction of Arrival Estimation (DOA), subspace vector, deconvolution algorithm

## Abstract

Utilizing the difference in phase and power spectrum between signals and noise, the estimation of direction of arrival (DOA) can be transferred to a spatial sample classification problem. The power ratio, namely signal-to-noise ratio (SNR), is highly required in most high-resolution beamforming methods so that high resolution and robustness are incompatible in a noisy background. Therefore, this paper proposes a Subspaces Deconvolution Vector (SDV) beamforming method to improve the robustness of a high-resolution DOA estimation. In a noisy environment, to handle the difficulty in separating signals from noise, we intend to initial beamforming value presets by incoherent eigenvalue in the frequency domain. The high resolution in the frequency domain guarantees the stability of the beamforming. By combining the robustness of conventional beamforming, the proposed method makes use of the subspace deconvolution vector to build a high-resolution beamforming process. The SDV method is aimed to obtain unitary frequency matrixes more stably and improve the accuracy of signal subspaces. The results of simulations and experiments show that when the input SNR is less than −27 dB, signals of decomposition differ unremarkably in the subspace while the SDV method can still obtain clear angles. In a marine background, this method works well in separating the noise and recruiting the characteristics of the signal into the DOA for subsequent processing.

## 1. Introduction

With the need for long-distance positioning increases, beamforming has become an efficient way to identify characteristic parameters of target signals, which has been widely applied in MIMO, 5G, and radar detection [1]. Classical beamforming methods, proved useful in practice, are based on “Fourier transform in direction domain” and can obtain high resolution on the promise that the input SNR is high enough or the datasheet is long enough. Among the beamforming methods, conventional beamforming (CBF) has lower resolution ratio but higher adaptability since CBF uses phase deviation without prior known number of sources, which is needed in high-resolution beamforming methods, such as Minimum Variance Distortion-less Response (MVDR) and Multiple Signal Classification (MUSIC) [2,3]. Recently, conversion methods of time-frequency are boosting, with better and faster performance to obtain frequency. The satisfying performance provides non-negligible opportunities for CBF in frequency domain to be both robust and high-resolution [4].

Time-frequency conversion is the key to handling the data in CBF of the frequency domain [5]. As a basic conversion in signal processing (SP), FFT has macroscopic effects in suppressing noise based on the randomness of noise [6,7]. The process of features extraction, however, works well in long data, resulting in a large amount of calculation. To discuss the effect of noise suppression, we assume that the number of arrays is M and the number of sampling points is N. It’s well-known that the spatial gain brought by the array is 10logM in the sonar equation. To make the incoming wave direction stand out in the beam diagram, namely, the incoming wave direction should be greater than the detection threshold (DT) after adding 10logM dB gain. When DT = 0, the maximum SNR recognized by conventional beamforming is −27 dB. In most cases, when the SNR is close to −27 dB, sometimes around −25 and −26 dB, some commonly used beamforming methods can no longer acquire the correct angle. Solving beamforming at extremely low SNR is a challenging problem.

To make full use of frequency information and overcome the complex calculation, beamforming has emerged in many high-resolution and effective methods. 

### 1.1. Conventional High Resolution Beamforming Methods

Basically, there are three kinds of algorithms. One is the subspace algorithm for separating signal and noise [8,9], one is the rotation-invariant algorithm for uniform linear array (ULA) [10,11], and the other one is the adaptive algorithm for changing backgrounds [12,13]. The common feature of the three algorithms is that the noise can be suppressed to the greatest extent so that they can obtain high resolution. Moreover, the Compressive Sensing (CS) theory and the Finite rate of innovation (FRI) theory, which have the superiority to undersample over CBF, have been applied in high-resolution beamforming estimation [14] and gain a lot of attention. They claimed to be sub-Nyquist sampled [15], but they require both a large amount of calculation and sparsity or degree of freedom for signals [16]. Compared with CBF, the methods mentioned above have achieved high-resolution results. However, they cannot acquire a satisfying result as CBF can under a low SNR input since noise correlation is usually generated during the operation of the system when there are multiple noises in the propagation. In addition, the sensors are easily triggered by multiple signals due to the distribution complexity of distances between different sources and backgrounds. Therefore, CBF still stands out in most practical applications. It’s challenging to combine high-resolution algorithms to suppress noise as well as the adaptability of CBF synchronously. 

### 1.2. How to Suppress Noise in Beamforming?

A large amount of research work has focused on noise reduction in beamforming and made great progress. P. Stoica et al. used an adaptive filter to reduce noise in the received signals [17]. The adaptive filter needed prior information about the environment around, which is also challenging to construct due to the mixed combination of sources and noise [18,19]. Wei Li et al. [20] used variational mode decomposition (VMD) to process array signals aiming to reduce some effects. In this way, the author claimed, a signal composed by multiple sources can be decomposed into a series of quasi-orthogonal intrinsic mode functions (IMFs) non-recursively [21]. Striving to reduce noise more accurately, Huang, YY et al. introduced a novel control parameter to make a tradeoff between the degree of source cross-correlation suppression and the violation of convex geometry under the promise of blind source separation [22]. In his research, the pre-processing approach showed less mean-square error (MSE) under the condition that input SNR is higher than 0 dB, which is far from a low SNR environment.

Some scholars tried to separate the vector space spanned by the signal and the noise without reducing the amplitude of noise, which is typical in beamforming algorithm: MUSIC and MVDR. Mats Viberg et al. had concluded sensor array processing algorithm based on subspace fitting [23]. They pointed out that this kind of algorithms was a maximum likelihood method (ML), and asymptotic properties in the maximum likelihood estimation can be achieved by introducing a special weight matrix [24].

Aiming at characteristics of the sources, many researchers have carried out related research concerned about networks. For example, Wang, SN et al. introduced an attribute-based double constraint denoising network (Att-DCDN), and Chen, Y et al. proposed a filter to ensemble the noisy signal and the reference one based on a backpropagation neural network (BPNN) [25,26,27]. Although these deep learning methods can reduce computation time to some extent, there is no engineering for exploring very low SNR cases.

### 1.3. How Does Deconvolution Decrease Noise?

Recently, T. C. Yang introduced the Richardson-Lucy deconvolution algorithm (RL) for high-resolution power spectral estimation [28]. Deconvolution, in fact, is suitable and convenient for the complex propagation process and is widely used in geological mapping [29]. Furtherly, the accurate identifications of target by deconvolution require two conditions: one is that all data must be nonnegative, and the other one is that target waves should integrate to 1 [30]. DOA matches the conditions fitly. Based on acoustic emission, the information of frequency and power obtained by the sensor array holds the potential of locating angles of acoustic emission using RL algorithm [31,32].

Considering the effect of deconvolution on image restoration, ref. [33] proposed to use multipath transmission model to transform data into frequency domain for deconvolution. In this way, time delay is mapped to phase shift in frequency domain to predict DOA and TDOA. They are aiming to predict DOA more accurately in multipath rather than recovery of lower SNR data, so they only tested data above 0 db. Uniquely in [34], deconvolution was used in the process of predicting noise distribution. Based on long-term observation, they summarized a class of heteroscedastic noise and applied deconvolution to the noise to estimate DOA in sparse Bayesian Learning (SBL). Ref. [35] also analyzed that the noise brought by deconvolution processing would affect computing ability to a certain extent.

Deconvolution is also widely used to improve resolution. Both [36,37] explored how to improve resolution. Ref. [36] considers the case that acoustic signals receive signals at a lower frequency than the designed way. They proposed a differential beamforming method and combined it with deconvolution to reduce the width of the beam power. However, in the test, the lowest SNR data they simulated was only 0 dB. Refs. [38,39] reconstructed to understand how convolution is applied in two-dimensional space circular array, and a clearer two-dimensional beam diagram was obtained.

Considering the iterative process in deconvolution, some studies have tested the time of deconvolution and proposed some algorithms for accelerating deconvolution. In [40], the author approximates the power propagation matrix to a symmetric STBT matrix, and deduces the regularized deconvolution algorithm of the convolution kernel for convolution. They reduced the computational complexity from $N_{m}\left(4N^{2}+3N\right)$ to $N_{m}\left(2N^{2}-2N\right)$. Other studies put forward a neural network to understand convolution for deconvolution in [41].

### 1.4. Contributions

The main contributions of this paper are summarized as the following points. Firstly, an optimized beamforming method named SDV is proposed. The method consists of two parts: a subspace deconvolution preprocessing for received signals in the frequency domain to obtain higher resolution frequency values, which provides accuracy for beamforming; A High-resolution beamforming based on the phase shift of signals after pre-processing. Secondly, the optimality and complexity of the proposed method are analyzed and compared with traditional beamforming methods. Thirdly, as a result, the proposed method achieves a noticeable improvement by comparing with other new beamforming methods. Results show that the proposed method can greatly improve the frequency resolution of signals and suppress noise to 20 dB even at a −27 dB-input of SNR. In beamforming figures, it is shown a more stable result with 1 Hz resolution, and 50 dB SNR improved.

The rest of paper is organized as follows: In Section 2, the principles and models of subspace deconvolution method of beamforming in the frequency domain are clarified; Section 3 discusses the effectiveness of proposed method and in this section, the signal enhancement effects of the method under different conditions are studied and plotted; In Section 4, the applicability of proposed method is verified by the beamforming data collected in the ocean; Finally, a summary is concluded in Section 5.

## 2. Problem Formulation and Proposed Method

### 2.1. Classical Method for CBF in Frequency Domain

Classical beamforming in the frequency domain, using Fourier series and conventional beamforming, have been applied in systems with considerable input. Based on the way signal comes, receivers acquire signal from:(1)$$J=A*S+N_{s}$$
where A is the steering vector-matrix with M rows and N columns; S is the source signal with N length and $N_{s}$ represents noise in the background.$\;N_{s}$ shares the same size with A. In most cases, $N_{s}$ is larger than A∗S in time domain. To remove the drawbacks $N_{s}$ matrix brings, Fourier series are used to highlight frequency feature in Equation (2):(2)$$X_{J}\left(k\right)=\overset{N-1}{\underset{n=0}{\sum }}J\left(n\right)e^{-j\left(2\pi /N\right)\mathrm{kn}}$$

For signals with single frequency, time delay can be written as:(3)$$e^{\mathrm{jw}\left(t_{0}+\tau_{n}\right)}=e^{{\mathrm{jwt}}_{0}}e^{{\mathrm{jw}\tau }_{n}}$$
where $\tau_{n}$ is the relative delay from the Nth receiver to the first receiver:(4)$$\tau_{n}={\Delta d}_{n}\sin\left(\theta \right)/c$$
where c is the wave speed and ${\Delta d}_{n}$ means the distance from n-th sensor to the first one. The Fourier transform is expanded into the superposition of single frequency signals, and then different phase shifts are added to different single frequency signals to obtain the final directivity.

### 2.2. Proposed SDV Method

Based on the conventional formula in Equation (1), autocorrelation is introduced to obtain signal subspaces:(5)$$R_{J}=E\left[JJ^{H}\right]$$
where H is Hermitian equivalent matrix. In most cases, the maximum likelihood estimation of the received data is usually used to remove $R_{J}$ above.
(6)$$\hat{R_{J}}=E\left[\left(AS+N\right)\cdot \left(AS+N\right)\right]=AR_{J}A^{H}+R_{N}$$
where $R_{N}$ denotes noise correlation matrix and can be expressed by:(7)$$R_{N}=\sigma^{2}I$$

There are usually D array elements in the array. In Equation (7), I is a unit matrix for the antenna array elements D·D in a practical scenario and $\sigma^{2}$ represents the noise amplitudes. The information of the most sources is able to be filtered out in this way. Consequently, the weaker one hidden in signals can be highlighted.

When decomposing the covariance matrix, the larger eigenvalues F and the rest of the smaller characteristic value can be obtained. The larger eigenvalue is more than eight times the order of magnitude of other eigenvalues so signal powers can be collected intensively. The eigenvectors corresponding to larger eigenvalues represent the signal subspace spanned by the signal vector, while the eigenvectors corresponding to other smaller eigenvalues represent the noise subspace spanned by the noise vector in a physical sense. Therefore, the result of decomposition can be expressed as:(8)$$R_{J}=Q_{S}\Sigma Q_{S}^{H}+Q_{N}\Sigma Q_{N}^{H}$$

Thus the $R_{J}$ in Equation (8) can be decomposed next:(9)$$R_{x}v_{i}=\lambda_{i}v_{i}$$
where $\lambda_{i}$ is the ith Eigenvalue of the matrix $R_{x}$, $v_{i}$ is the eigenvector corresponding to $\lambda_{i}$ with a M length.

Considering the noise as white Gaussian noise:(10)$$\lambda_{i}=\sigma^{2}$$

We can conclude that:(11)$$R_{x}v_{i}=\lambda_{i}v_{i}=\sigma^{2}v_{i}=\left(AR_{S}A^{H}+\sigma^{2}I\right)v_{i}$$
(12)$$AR_{S}A^{H}v_{i}=0$$

In Equation (13), $A^{H}A$ is full matrix of dimension D, so that ${\left(A^{H}A\right)}^{-1}$ and $R_{S}^{-1}$ exist as $R_{S}$ is ideal source signal.
(13)$$R_{S}^{-1}\left(A^{H}A\right)A^{H}AR_{S}A^{H}v_{i}=0$$
(14)$$A^{H}v_{i}=0,i=D+1,i=D+2,\ldots M$$

Equation (14) shows the eigenvector corresponding to the noise eigenvalue is orthogonal to the column vector of matrix **A**, and each column of **A** corresponds to the direction of the signal source, which is the starting point for solving the direction of the signal source by using the noise eigenvector. Based on the orthogonality of signal subspace and noise subspace, the spatial-spectral function is constructed, and parameters of the signal are estimated by searching the spectral peak. After the subspace decomposition, the RL deconvolution is applied to obtain higher frequencies.

For the received data p(x) in Equation (1), the blurring process can be represented as the convolution of the clear signal and a Point Spread function (PSF) plus noise:(15)P=I⊗K+N
where ⊗ means a Kronecker product. P is the blurred data, which replaces J as mentioned above; I represents the desired signal, which corresponds to the ideal signal in the signal process; K is PSF with the same size with A; N is white Gaussian noise with M rows and N columns, respectively. For Poisson noise distribution, likelihood probability of the desired I can be expressed as:(16)$$p\left(P|I\right)P=\overset{N}{\underset{x}{\prod }}\left(I⊗K\right)x^{B\left(x\right)}\exp\left[-\left(I⊗K\right)x\right]B\left(x\right)!$$
(17)$$B_{P}\left(x\right)=\mathrm{Poisson}\left(\left(I⊗K\right)\left(x\right)\right)$$

To acquire solutions of the max likelihood probability, the power function needs to be minimized:(18)$$I^{*}=\mathrm{argmin}\;E\left(I\right)$$
(19)E(I)=∑[(I⊗K)−P×log(I⊗K)]

Equation (19) is called Kullback-Leibler Divergence, also known as relative entropy, so as to quantify the resemblance on two different probability distributions. The K-L divergence helps us measure the loss of information when using one distribution to approximate another. There are two prerequisites for the Lucy–Richardson algorithm: received data, PSF, and ideal data must be nonnegative; the first two data must integrate to 1. The prerequisites imply that the RL algorithm has two characteristics: nonnegative, which guarantees the estimated values are all positive; Energy retention, which maintains full energy during the iteration. By controlling constraints and derivative of the preconditions, iteration of the solution can be expressed as:(20)$$I^{t+1}=I^{t}\left[K^{\frac{P}{I^{t}⊗K}}\right]$$
where t is the number of iteration, and it varies with different inputs. For the original discrete Fourier transform, discrete signals of finite length can be expressed as:(21)$$X\left(k\right)=\overset{N-1}{\underset{n=0}{\sum }}x\left(n\right)W_{N}^{\mathrm{kn}},k=0,1,\ldots N-1$$
(22)$$W_{N}^{\mathrm{kn}}=e^{-j\frac{2\pi }{N}}$$

Compared with the RL deconvolution, the PSF can be replaced by:(23)$$X_{k}=\overset{L}{\underset{i=1}{\sum }}{\left|{\left(W_{N}^{\mathrm{kn}}\right)}^{T}v_{j}W_{N}^{\mathrm{kn}}\right|}^{2}$$
where L is the length of data.

Subspace deconvolution is applied to filter most original data from the background as it is similar to a process to acquire a clear image from a blurred one. After we acquire a more accurate subspace vector, beamforming can be performed based on phase shift, which is classical in CBF in Equation (2).
(24)$$P_{k}=\overset{M}{\underset{i=1}{\sum }}X_{k}w^{H}A$$

To acquire the direction of the incoming wave, the Plancherel theorem is applied to acquire the most powerful direction of all sensors in Equation (24). In Equation (24), $P_{k}$ is the power of beamforming based on different theta, which is usually from $90^{\circ }$ to $-90^{\circ }$. **w** represents the weight vector of the beamforming and **A** corresponds to Equation (2). The flow chart of the SDV method is shown in Figure 1. The performance of the proposed SDV method is discussed later with simulated and measured data.

### 2.3. Performance of SDV Method

Improvement of SNR and computation loads of SDV method are discussed below.

For the m-th receiver, the received signal is:(25)$$x_{m}\left(t\right)=\overset{i=M}{\underset{i=1}{\sum }}A_{i}e^{j2{\pi f}_{i}\left(t+\tau_{m}\right)}+n_{m}\left(t\right)$$
where $\tau_{m}$ represents the delay relative to the first sensor and **n**(t) represents the noise received by the sensor.

Generally, we perform FFT transformation on the received data **X** of the whole arrays to obtain its Fourier amplitude spectrum, and then carry out phase superposition for the spectrum. In the Frequency domain, Equation (25) can be expressed by Equation (26). In this way, the gain of the entire output is 10logM (M is the number of arrays).
(26)$$AF\left(f\right)=AF_{S}\left(f\right)+AF_{N}\left(f\right)$$

While in our proposed pre-processing, expectation of **AF** is:(27)$$E\left[AF\left(f\right)\right]=\overset{m=\left(L-1\right)}{\underset{m=-\left(L-1\right)}{\sum }}E\left[R_{SDV}\left(t\right)\right]e^{-j2\pi \mathrm{fm}}=AF_{SDV}\left(f\right)⊗AF_{W}\left(f\right)$$
where N states the number of sampling and **R_{SDV}** is a Toeplitz matrix after subspaces decomposition. Therefore, Equation (26) can be written by:(28)$$AF\left(f\right)=AF_{S}\left(f\right)⊗AF_{W}\left(f\right)+AF_{N}\left(f\right)⊗AF_{W}\left(f\right)$$

Considering a regular Fourier transform:(29)$$W\left(k\right)=e^{-j\left(N-1\right)\pi k}N\frac{\sin c\left(\pi \mathrm{Nk}\right)}{\sin c\left(\pi k\right)}$$
(30)$$AF_{w}=N\frac{\sin c\left(\pi \mathrm{Nk}\right)}{\sin c\left(\pi k\right)}$$

For data with a N-points length, its SNR before and after transformation is defined as:(31)$${\mathrm{SNR}}_{N}=\frac{AF_{SW}\left(f\right)}{AF_{NW}\left(f\right)}$$

We assume that there is only one simple sine function in source: $x_{S}=\mathrm{Asin}\left(2{\mathrm{pif}}_{0}t+\phi \right)+n\left(t\right)$ and the noise consists of Gaussian noise. In addition, the original sine distribution in the frequency domain is:(32)$$AF_{S}\left(f\right)=A\left[\delta \left(f-f_{0}\right)+\delta \left(f+f_{0}\right)\right]$$
(33)$$AF_{N}\left(f\right)=\chi_{0}$$
where $\chi_{0}$ represents a distribution of white noise in the frequency domain.

After the SDV:(34)$$AF_{xm}\left(f\right)=AF_{S}\left(f\right)⊗AF_{w}^{SDV}\left(f\right)+AF_{N}\left(f\right)⊗AF_{w}^{SDV}\left(f\right)=AF_{SW}^{SDV}\left(f\right)+AF_{NW}^{SDV}\left(f\right)$$
(35)$${\mathrm{SNR}}_{\mathrm{SDV}}=\frac{AF_{SW}^{SDV}\left(f\right)}{AF_{NW}^{SDV}\left(f\right)}$$

To further analyze the gain:(36)$${\mathrm{SNR}}_{N}=\frac{AF_{S}\left(f\right)⊗AF_{w}\left(f\right)}{AF_{N}\left(f\right)⊗AF_{w}\left(f\right)}=\frac{A\delta \left(f-f_{0}\right)⊗AF_{W}\left(f\right)}{\chi_{0}⊗AF_{W}\left(f\right)}$$
(37)$${\mathrm{SNR}}_{\mathrm{SDV}}=\frac{AF_{S}\left(f\right)⊗AF_{w}^{SDV}\left(f\right)}{AF_{N}\left(f\right)⊗AF_{w}^{SDV}\left(f\right)}=\frac{A\delta \left(f-f_{0}\right)⊗AF_{W}^{SVD}\left(f\right)}{\chi_{0}⊗AF_{W}^{SVD}\left(f\right)}$$

The improvement of SNR is:(38)$$\mathrm{Gain}=10\log\left(\frac{{\mathrm{SNR}}_{\mathrm{SDV}}}{{\mathrm{SNR}}_{N}}\right)=10\log\left[\frac{\chi_{0}⊗AF_{W}^{SVD}\left(f\right)}{\chi_{0}⊗AF_{W}\left(f\right)}\right]$$

Based on the Fourier transform, the frequency resolution is $a=\frac{\mathrm{fs}}{N}$. Therefore, Equation (38) can be written as:(39)$$\mathrm{Gain}=10\log\left[\frac{\sum_{k=f_{0}-\mathrm{ka}}^{k=f_{0}+\mathrm{ka}}\chi_{0}AF_{w}\left(f_{0}-k\right)}{\sum_{k=f_{0}-\mathrm{ka}}^{k=f_{0}+\mathrm{ka}}\chi_{0}AF_{w}^{SVD}\left(f_{0}-k\right)}\right]$$

The gain of pre-processing is based on the gain of frequency domain transformation, and its calculation effect is related to the number of iterations. Perfect deconvolution can achieve a δ(f), which effectively inhibits spectrum leakage. In the specific SDV processing, Equation (25) can be expressed by:(40)$$X=cx+\sigma_{n}^{2}b$$
(41)$$X_{P}=w_{n}^{H}E\left[XX^{H}\right]w_{n}=w_{n}^{H}\frac{1}{N}\overset{M}{\underset{i=1}{\sum }}X_{i}X_{i}^{H}$$
(42)$$w_{n}=\frac{a_{n}}{{\left|\left|a\right|\right|}_{2}^{2}}$$
(43)$$C=\left[\begin{array}{ccc} \frac{|{\left|a_{1}^{H}a_{1}\right||}_{2}^{2}}{{\left|\left|a_{1}\right|\right|}_{2}^{4}} & \cdots & \frac{||{a_{1}^{H}a_{N}||}_{2}^{2}}{{\left|\left|a_{1}\right|\right|}_{2}^{4}}\\ \vdots & \ddots & \vdots \\ \frac{|{\left|a_{N}^{H}a_{1}\right||}_{2}^{2}}{{\left|\left|a_{N}\right|\right|}_{2}^{4}} & \cdots & \frac{||{a_{N}^{H}a_{N}||}_{2}^{2}}{{\left|\left|a_{N}\right|\right|}_{2}^{4}} \end{array}\right],\frac{||{a_{1}^{H}a_{1}||}_{2}^{2}}{{\left|\left|a_{1}\right|\right|}_{2}^{4}}={\left|\overset{M}{\underset{m=1}{\sum }}X_{\text{P}}{\text{e}}^{-j2\pi /N}\right|}^{2}$$

Equation (43) is the PSF in Equation (15). So Equation (40) can be expressed by an optimization problem:(44)$$x=\mathrm{argmin}{\left|\left|cx-y\right|\right|}_{2}^{2}+\gamma g\left(x\right)$$

And the gradient of Equation (44) is:(45)$$\nabla J\left(x\right)=2H^{H}\left(cx-y\right)+\gamma \frac{\partial g\left(x\right)}{\partial x}$$

So that:(46)$$x^{i+1}=x^{i}-u\nabla J\left(x\right)$$

It is worth noting that because of the high-resolution recognition in the frequency domain in the previous step, $\sigma_{n}^{2}b$ in Equation (40) has been greatly reduced to almost zero. Perform beamforming and deconvolution transformation on the above equation, and replace Equation (43) with:(47)$$\frac{||{a_{1}^{H}a_{1}||}_{2}^{2}}{{\left|\left|a_{1}\right|\right|}_{2}^{4}}={\left|\frac{1}{\sum_{m=1}^{M}\frac{1}{d_{i,m}^{2}}}\overset{M}{\underset{m=1}{\sum }}\frac{1}{d_{i,m}d_{j,m}}e^{-j2\pi f_{i}{\left(\tau_{j,m}-\tau_{i,m}\right)}^{}}\right|}^{2}$$

According to Equation (43), this calculation includes a matrix decomposition process and an iterative deconvolution process, and its computational complexity is $M\left(4N^{2}+3N\right)$ [40].

In the test, the gain obtained by this method can still be close to 18logN at low SNR. Through the analysis, we can conclude that pre-processing uses subspace deconvolution to improve the resolution and gain of signals in the frequency domain. In fact, the gain improvement is reflected in the stability and adaptability of the algorithm. That is, this stage is to prepare the data for the subsequent beamforming stage. Other algorithms mentioned in related work are unable to achieve the ideal gain curve due to the fact that some false peaks appear in noise separation and the gain cannot be accurately added when the SNR is low. This method can reduce the pseudo peak in this separation stage, which means that the method can not only maintain the high resolution, but also reduce the high resolution used in the “pseudo peak” by other high-resolution algorithms by inputting a frequency domain with less interference into beamforming and then shrinking the beam width.

## 3. Performance of SDV Method in Simulations

To formulate the problem, we have carried out numerical simulations based on an ideal Gaussian noise environment as well as practical experiments under a physical seabed environment are performed to evaluate the performance of the proposed.

### 3.1. Performance of Frequency Resolution

#### 3.1.1. Noise Suppression

Specified frequency periodic signals are set up as 50 Hz, and the sampling frequency is set up as 1 kHz. We consider a signal with center frequency = 50 Hz as presented above with random white noise. In Figure 2, the curve, denoted by “SDV”, is compared with the classical FFT results and the “DCV” in [2] with an input SNR of −10 dB and −20 dB, respectively.

Figure 2 illustrates that the proposed pre-process method improves the output SNR by 20 dB roughly and that all corresponding noise is suppressed. It’s worth noting that in the case of input SNR = −10 dB and −20 dB, the proposed method does not suppress the single frequency signals that may exist in the noise. For example, the signals are prominent around 50 Hz and in 400 Hz in Figure 2, which corresponds to the results of FFT. When SNR = −20 dB, we can highlight the 300 Hz signal so that other information about the environment will not be ignored. In practice, when there is a fixed interference frequency in the background, the proposed method is more meaningful than FFT. Moreover, in FFT transformation, when the input SNR is reduced by 10 dB, the spectrum produced by FFT also varies by about 10 dB, while the spectrum produced by the proposed method is less than 5 dB, which indicates the method also gives rise to the signal.

#### 3.1.2. Influence of Data Length on SDV Method

In this simulation, in view of the fact that the effects of the SDV method are prominent for any length of data, the influence of data length on proposed pre-processing method is discussed. As shown in Figure 3, we set the center frequency as 50 Hz and set different sampling duration as 3000 in Figure 3a and 5000 in Figure 3b. Compared with each figure, it can be explicitly seen that in addition to the signal gain brought by deconvolution, the time gain is increased by 5 dB, which is consistent with the theoretical value. Furthermore, in addition to the higher signal resolution, the noise around 250~300 Hz is better suppressed due to the lower correlation of noise when sampling time grows longer.

#### 3.1.3. Influence of SNR on SDV Method

As stated above, a frequency spectrum estimated by the proposed method is obtained by deconvolving data calculated by the conventional method. The high suppression of noise obtained through proposed method depends on specifying target frequency from noise, and it affects beamforming in the next step seriously. Therefore, it is necessary to analyze the influence of input SNR on the pre-process.

In this simulation, experiments use a group of signals with the same sampling rate, center frequency sinusoidal, but different input SNR to demonstrate our method. These signals are superimposed in the Gaussian noise, with SNR valuing from −26 dB to −20 dB. Due to the randomness of the Gaussian white noise, each SNR value set 10 groups to count the accuracy of decomposition on signal characteristics. Mean values of SNR promoted by SDV method are compared with FFT in Table 1, which is an input for beamforming. Some interesting figures showing how the pre-process affects spectrum significantly have been drawn in Figure 4 and Figure 5, respectively.

In Table 1 and Figure 4, it can be found that in the traditional FFT, when the SNR of the input signal increases, the SNR of the output almost increases with the same difference. While the proposed pre-process shows improvement of more than 20 dB on most occasions. In addition, the resolution, as well as narrow beamwidth remain stable in Table 1. To discuss the performance of proposed method, a clear improvement of SNR is calculated. Since subspaces of signals decomposed by the pre-process mean less influence of noise, the output SNR is a direct indication showing that the proposed method will not be affected by the input SNR within a certain range and its resolution is stable at 1 Hz when SNR is relatively small.

The same result is particularly evident in Figure 5. With the decrease of SNR value, the curve of FFT gradually approaches 0 on the coordinate axis, and the signal is also gradually submerged and cannot be seen. The curves of (a) (b) and (c) in Figure 5 are very similar. The effectiveness of the algorithm is more obvious compared to Figure 4d and Figure 5. The center frequency calculated by FFT has been almost submerged by noise and cannot be seen, but our algorithm can still excellently highlight the target signal, and the frequency resolution remains 1 Hz when the SNR is −26 dB. More details can be found in Table 1. Thus, we may conclude that our pre-process can highly give rise to gains of signals and provide better noise suppression as well as high resolution.

### 3.2. Performance of DOA

In this simulation, our algorithm is compared with three traditional but widely used algorithms and five new high-resolution algorithms. The traditional algorithm contains: CBF, MUSIC and Compressive Sensing(CS) [42,43]. in addition, the new high-resolution algorithm contains: Improved-L1-SVD [44], Improved-4order-MUSIC [45], Weighted Spatial Covariance Matrix MUSIC (WSCM-MUSIC) [46], Model-Based Expectation Maximization Beamforming (MESSL) [47] and Maximum-correlation with Simplex geometry Beamforming (MS-PHAT) [48]. In the simulation, Orthogonal Matching Pursuit (OMP) is applied for the data recovery process in the CS.

#### 3.2.1. Performance of High Resolution

The simulation signal involves signals whose directions are $\mathrm{angle}=33^{\circ }$, and the data length is T = 3 s with a sampling frequency of 10 kHz. SNRs are set from 20 dB to −20 dB. Some representative pictures are shown in Figure 6, Figure 7 and Figure 8. In this part, TIC and TOC are used to count the execution CPU time in MATLAB. We choose DELL OptiPlex 7070 as a convenient platform, which has a modest CPU (3.00 GHz Intel Core i7-9700) and a moderate memory space (64.0 GB RAM) for data processing.

In Figure 6, CBF works as a contrast with the other methods. Due to the high power of the strong source signal and high array gain, we can detect a correct direction with the highest side lobes in CBF and similar side lobes in CS and Im-L1-SVD. In the case of high SNR, the resolution of the proposed SDV method is less than 1° and there is almost no side lobe. CS, working well in high resolution, however, detects some pseudo directions. When the noise becomes stronger, it has the greatest influence on other algorithm because the interference causes instability to the optimization algorithm. Namely, mismatch problems may happen and have a strong impact on DOA estimation, especially for weak signals.

In Table 1, compared with other methods, the proposed SDV method has a consistent improvement on frequency resolution without impacting SNR improvements. In addition, SDV has the ability to suppress sidelobes in the best possible manner, as is evident in Figure 5a, where sidelobes are suppressed completely, the same as the result in Figure 5b. At the same time, SDV brings no other extra angles that may be brought in data reconstruction by local minimum estimation error in CS, as is evident in Figure 6d. Although CS and Im-L1-svd can bring great improvement in SNR development, wrong directions can be produced at the same time due to the low SNR of input. It is worth noticing that grid mismatch and the large number of grids used in CS can lead to deteriorated recovery performance [28]. Taking accuracy and resolution into account, the proposed SDV method can provide the most satisfying estimation results with the least calculation. Furthermore, SDV possesses the added characteristic to catch weak signals.

#### 3.2.2. Less Sensibility to Noise

For each SNR value, we conducted 10 experiments with different methods. In order to demonstrate the advantages of our algorithm, we focus on the case of negative SNR. RMSE of all methods with the input SNR of −10 dB are plotted. As simulated above, due to the capacity to suppress sidelobes and acquire a high direction resolution, we conclude that the SDV method is suitable to detect weak signals. 

Im-L1-svd method is also based on noise subspace, and it is also obvious that this algorithm has a good suppression effect on stray peaks. However, in this experiment, symmetric peaks appear, because only subspace separation with superimposed weights [44] is used, which will lose part of information due to energy leakage. This is not good for our actual positioning, although the resolution of the method is very impressive.

MUSIC algorithm is very similar to the result of Im-4order-Music, as the main principle of the two algorithms is the same, both of them come from the direct subspace decomposition of array data, and Im-4order-Music algorithm also uses 4 higher-order spectrums to suppress noise. Therefore, in this experiment, the effect of Im-4order-Music algorithm is better than that of MUSIC algorithm. It is worth mentioning that Kronecker product, corresponding to kron function in MATLAB, is used in Im-4order-Music for fourth-order calculation. It is this limitation that results in a very large memory footprint, exceeding the memory capacity of most computers when the number of elements exceeds a certain number.

In the comparative experiment, the resolution and improved SNR could not be seen by three methods. MESSL uses the probability model that predicts the distribution of sound sources to predict the location of the sound source, and estimates it through expectation maximization. Similar to the original literature, when the SNR is greater than 0, the prediction is more accurate. MS-PHAT algorithm uses a frame of data for calculation, so the calculation time of this method is the shortest, namely real-time, but its accuracy is not optimistic. Compared with the original MUSIC algorithm, WSCM-Music uses time-frequency transformation to calculate the weighted value. This weighting comes from a long Short-Term memory network that takes a time-frequency graph as input and a binaries time-frequency graph as output. In [46], even at 20 dB, the time-frequency diagram output by the network only contains a small piece of information, and most of the signal information is set to 0 due to noise, so there is only one angle output. This conversion network is not suitable for most cases, especially low SNR cases. This also results in the highest RMSE for this method.

As shown in Figure 7 and Figure 8, the DOA spectrum of CBF shows a poor resolution under the SNR = −10 dB and can hardly detect the direction when the SNR value is −20 dB. MUSIC detects target direction with much lower resolution and more interference compared with Figure 6. Remaining to be both high-resolution, as CS shows in both figures, however, low SNR value obstructs accuracy of OMP method. Through SDV, the accurate direction can be detected with high-frequency resolution. In addition, within the observation range of the figure, sidelobes are almost completely suppressed. 

More details of direction estimation obtained through methods mentioned above are shown in Table 2 with an input SNR = −27 dB, which is emphasized in our proposal. Under the circumstance of intense noise interference, it is hard to detect the weak line signal. As is shown in both figures and table, SDV holds a high resolution even when the SNR is as low as −27 dB and has a powerful ability to highlight signals. This makes sure that those weak signals will not be engulfed by noise in background. Experiments data will be tested on the methods in the following part. In Figure 9, we can confirm the previous conclusion that CBF is the most used method at low SNR. However, our proposed method can be infinitely close to CBF in terms of stability, while maintaining high resolution.

## 4. SDV for Experimental Data Analysis

In this section, the abilities to detect weak signals affected by loud interference and comparison between other methods in a noisy background are analyzed using experimental data. Considering that other new algorithms have shown shortcomings in the simulation stage, especially the WSCM-MUSIC algorithm needs to train the network separately, conventional algorithms are used for comparison in the actual data test.

The experimental data used comes from channels of the VLA Array in SWellEx-96 Event S59. It contains a loud interferer along an isobath in the experiment in 1996. In the experiment, the deep source was towed at a depth of about 54 m. It transmitted numerous tonals of various source levels between 49 Hz and 400 Hz. We take two sets of 5000 snapshots of them and compare the differences between methods.

Figure 8 shows the frequencies transmitted from the experimental data with ocean background noise and the Bearing Time Recording (BTR) of different methods. Comparison between SDV and FFT show that DCV can provide higher frequency resolution and reduce the influence from sidelobes in Figure 10a,b, while pre-process of SDV provide a much higher noise suppression. The output SNR of Figure 8a reaches 38 dB higher than FFT and 28 dB higher than DCV by means of SDV method. Applying the frequency in beamforming, as shown in Figure 10c,d, the direction of arrivals is detected. Details of beamforming are illustrated in Table 3.

Focusing on the resolution first, CS performs best since the beam of angle acts as well as simulation. Remaining high resolution, however, CS acquires wrong directions. The advantages of the SDV method for frequency detection are evident through the suppression of background noise. Therefore, this method is suitable to estimate a high-resolution spectrum and detect feeble signals, which is easy to be applied in other SP areas.

## 5. Conclusions

In this paper, through the combination of subspaces and deconvolution, the resolution of direction is greatly improved, and the robustness of beamforming under a low-SNR environment can be ensured. Compared with other useful beamforming methods, the proposed SDV method can better preserve the signal subspace effectively, thus obtaining and keeping a higher resolution. Moreover, with the decrease of input SNR, SDV remains correct direction and appropriate calculation time. Above all, the satisfactory performance of the proposed SDV method is demonstrated both in proof of formula and in many simulations. Through the SDV method, the stronger noise can be filtered out, and weaker sources can be highlighted than other methods. Finally, the resolution of SDV remains 1 Hz and improved SNR remains higher than 50 dB in Marine background noise.

In the future, we will study the evolution process of acoustic emission and the optimization of iterative process to achieve better beamforming results.

## Figures and Tables

**Figure 1 sensors-22-02327-f001:**
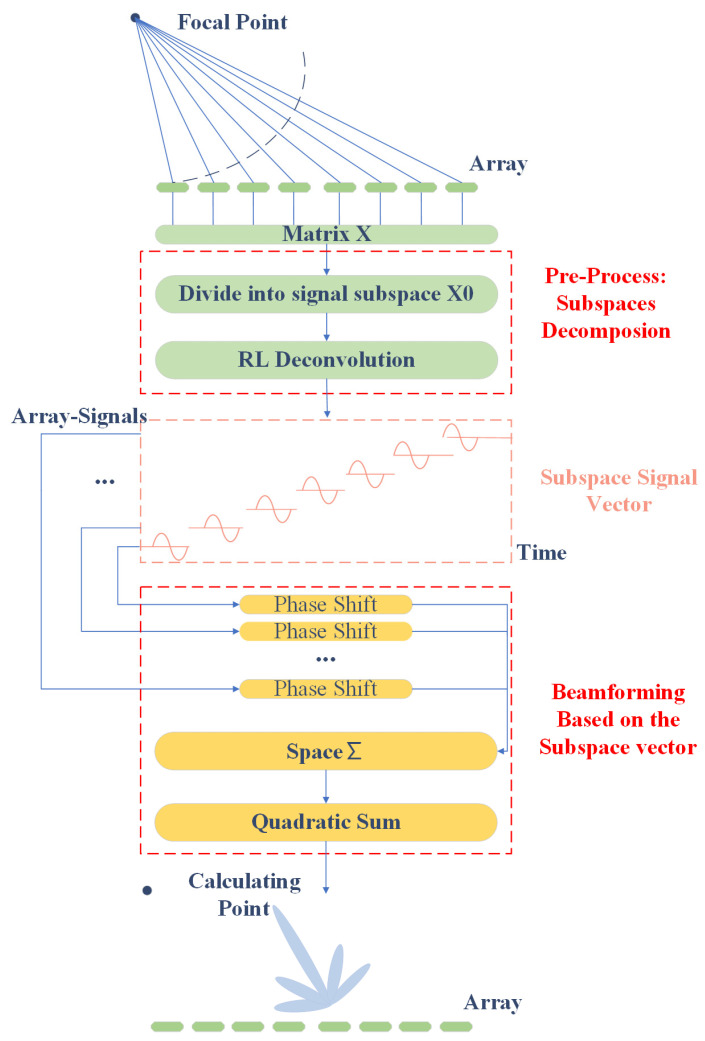
Flow chart of SDV method.

**Figure 2 sensors-22-02327-f002:**
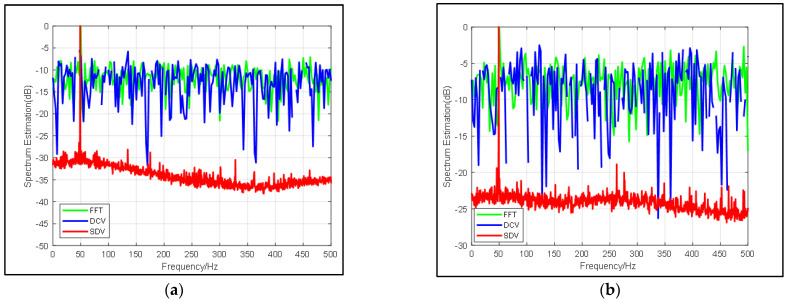
Comparison of frequency spectrum estimation results from FFT (blue line), DCV (green line) and SDV (red line) with (**a**) SNR = −10 dB and (**b**) SNR = −20 dB.

**Figure 3 sensors-22-02327-f003:**
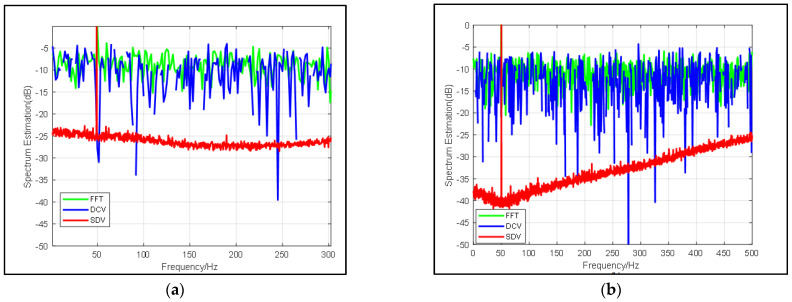
Comparison of frequency spectrum estimation results from FFT (blue line), DCV (green line) and SDV (red line) with same input SNR = −15 dB and different snapshots: (**a**) snapshots = 3000 and (**b**) snapshots = 5000.

**Figure 4 sensors-22-02327-f004:**
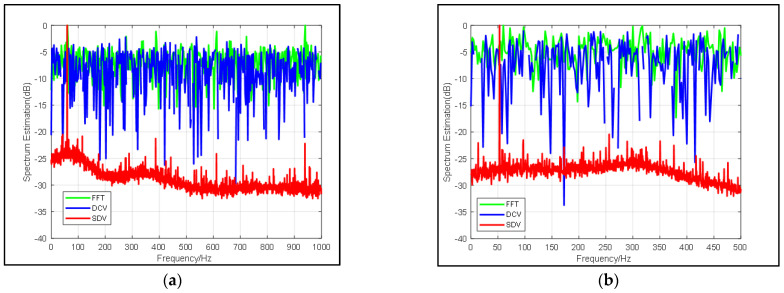
Comparison of frequency spectrum estimation results from FFT (blue line), DCV (green line) and SDV (red line) with same snapshots but different SNR: (**a**) SNR = −20 dB, (**b**) SNR = −22 dB, (**c**) SNR = −24 dB, (**d**) SNR = −26 dB.

**Figure 5 sensors-22-02327-f005:**
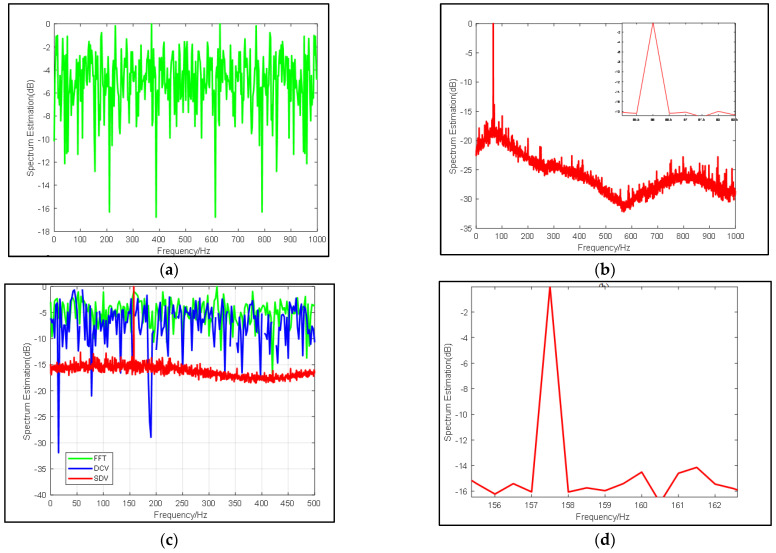
(**a**) Results from FFT under −26 dB, (**b**) Results from SDV using the same data with (**a**), (**c**) Comparison of frequency spectrum estimation results from FFT (blue line), DCV (green line) and SDV (red line) under SNR = −26 dB, (**d**) Zoom in on the spike of (**c**).

**Figure 6 sensors-22-02327-f006:**
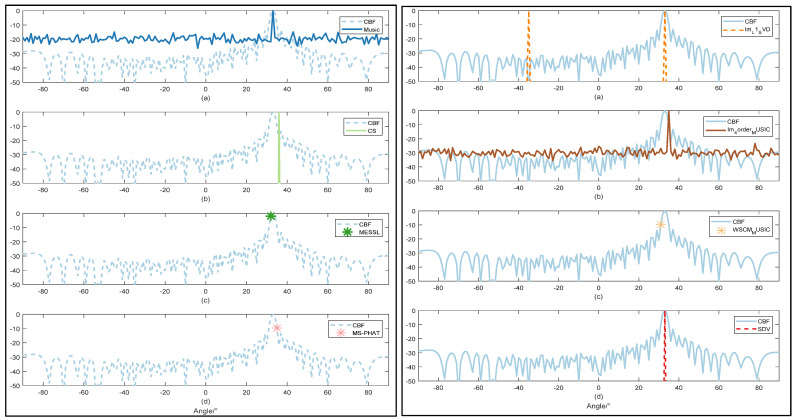
Comparison of direction estimation with SNR = 10 dB by 9 methods: CBF worked as a comparison in all figures. Music are shown in left (**a**) CS are shown in left (**b**). MESSL are shown in left (**c**) and MS-PHAT are shown in left (**d**). Im-L1-SVD are shown in right (**a**). Im-4order-MUSIC are shown in right (**b**). WSCM-MUSIC are shown in right (**c**) and proposed SDV are shown in right (**d**).

**Figure 7 sensors-22-02327-f007:**
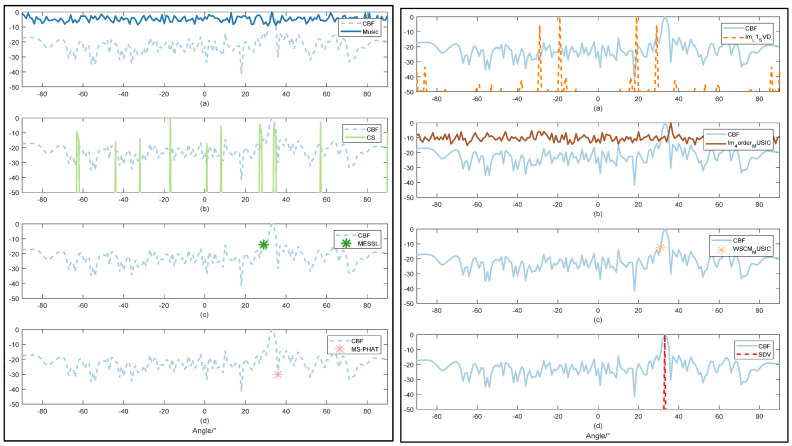
Comparison of direction estimation with ${\mathrm{SNR}}_{1}=-10\;\mathrm{dB}$ by 9 methods: CBF worked as a comparison in all figures. Music are shown in left (**a**) CS are shown in left (**b**). MESSL are shown in left (**c**) and MS-PHAT are shown in left (**d**). Im-L1-SVD are shown in right (**a**). Im-4order-MUSIC are shown in right (**b**). WSCM-MUSIC are shown in right (**c**) and proposed SDV are shown in right (**d**).

**Figure 8 sensors-22-02327-f008:**
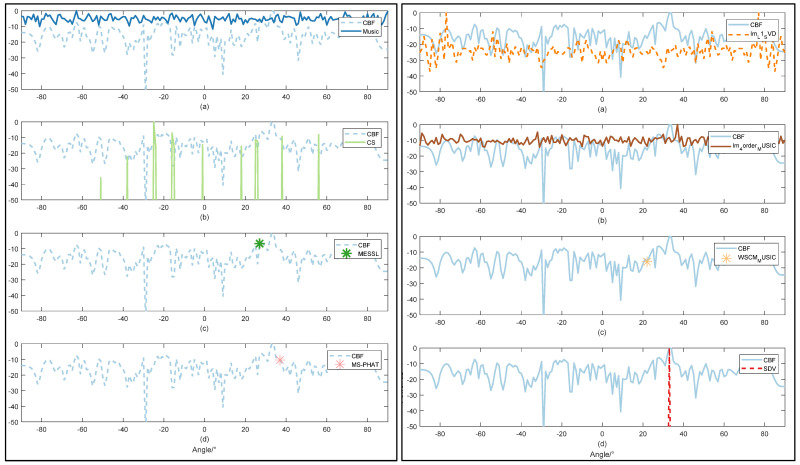
Comparison of direction estimation with ${\mathrm{SNR}}_{1}=-20\;\mathrm{dB}.$ by 9 methods: CBF worked as a comparison in all figures. Music are shown in left (**a**) CS are shown in left (**b**). MESSL are shown in left (**c**) and MS-PHAT are shown in left (**d**). Im-L1-SVD are shown in right (**a**). Im-4order-MUSIC are shown in right (**b**). WSCM-MUSIC are shown in right (**c**) and proposed SDV are shown in right (**d**).

**Figure 9 sensors-22-02327-f009:**
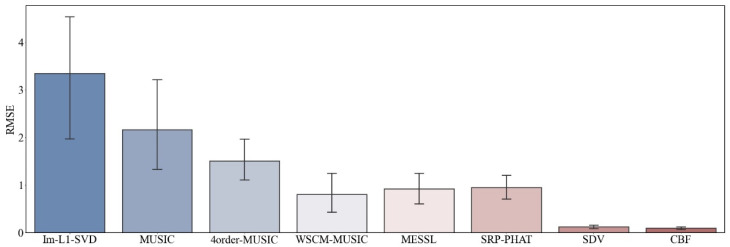
RMSE of direction estimation with ${\mathrm{SNR}}_{1}=-10\;\mathrm{dB}$.

**Figure 10 sensors-22-02327-f010:**
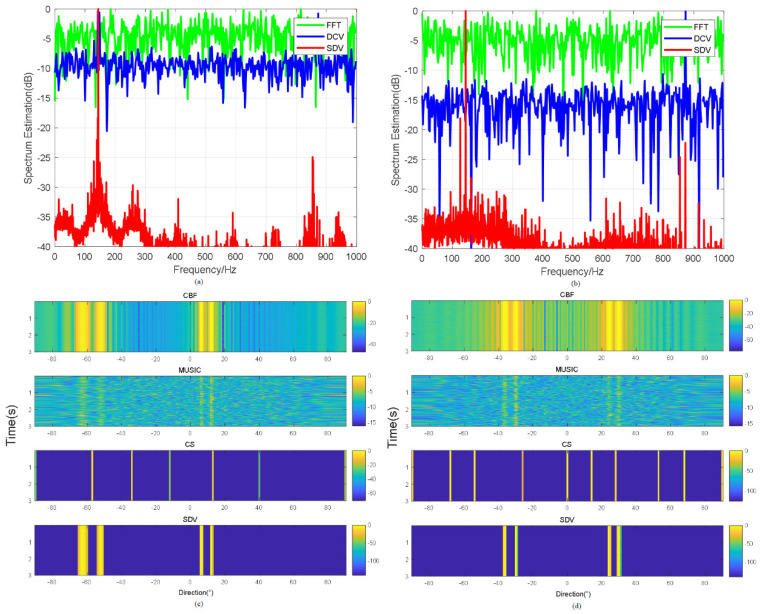
Comparison of FFT, DCV and proposed pre-process of SDV in first set of data as shown in (**a**) and the second set of data as shown in (**b**); Comparison of direction estimation obtained through CBF, MVDR, CS and SDV in experiment with the first set data: ${\mathrm{direction}}_{1}=-62^{\circ },-57^{\circ },7^{\circ },11^{\circ }$ as shown in (**c**) and the second set data:${\;\mathrm{direction}}_{1}=-48^{\circ },-30^{\circ },22^{\circ },31^{\circ }$ as shown in (**d**).

**Table 1 sensors-22-02327-t001:** Influence of input SNR on output SNR and width of frequency.

Input SNR (dB)	Output SNR (dB) ^1^	Output SNR (dB) ^2^	Beam Width (Hz) ^2^
−27	−2	16.3	1
−26	3	24.8	1
−25	3.5	25.4	1
−24	4	25.8	1
−23	4.5	26.9	1
−22	5	28.1	0.1
−21	5.5	29.3	<0.1
−20	6.5	30	<0.1

^1^ By means of conventional FFT. ^2^ By means of SDV method.

**Table 2 sensors-22-02327-t002:** Comparison of different beamforming methods with input SNR = −27 dB and $\mathrm{targetdirection}=33^{\circ }.$

Methods	CBF	MUSIC	CS-OMP	Im-L1-SVD	4order-MUSIC	WSCM-MUSIC	MESSL	SRP-PHAT	SDV
Direction (°)	−42, 10	9, 78	0, 17, 31	29	9, 78	−20, 46	42	0	33
SNR (dB)	10.45	11.87	>50	37.1	17.54	-	-	-	>50
Width (°)	>2	2	0.6	1.2	1.3	-	-	-	1.2
Time (s)	0.318	2.442	431.646	588.574	4.588	4.637	5.648	0.045	2.25

**Table 3 sensors-22-02327-t003:** Comparison of different beamforming methods under experimental data (a).

Methods	CBF	MUSIC	CS	SDV
Direction (°)	−62, −57, 7, 11	−62, −57, 7, 11	−59, −37, −17, 15, 40	−62, −57, 7, 11
SNR (dB)	20.1	17.4	>50	>50
Width (°)	2	2	0.7	1.2

## Data Availability

The data that support the findings of this study are available from the corresponding author upon reasonable request.

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
