# Peer review of "An Optimal Subspace Deconvolution Algorithm for Robust and High-Resolution Beamforming"

_sensors, 2022, doi:10.3390/s22062327_

Round 1
Reviewer 1 Report
This paper proposed an Optimal Subspace Deconvolution Algorithm for Robust and High-Resolution Beamforming. It is very interesting, but there still are some problems in this paper.
- More deconvolution methods should be reviewed rather than other methods, such as VMD.
- The SNR with -27 dB is highlighted in this paper. The special reason of -27 dB should be pointed.
- Other methods proposed recently should be used in this paper to highlight the advantages of the proposed method.
- The pictures of this paper should be clearer.
Author Response
Dear Reviewer,
We submitted a manuscript in Feb.2022, and the paper was recommended a major revision. Based on the helpful comments, we have revised the paper.
In the main text of the revised version, the main revisions are highlighted with red bold. Please see the attachment.
We sincerely hope the revised manuscript meets the requirements of the reviewers.
Best regards!
Xiruo Su
College of Biomedical Engineering & Instrument Science,
Zhejiang University, Hangzhou, 310058, China.
E-mail: xiruo_su@zju.edu.cn

Reviewer 2 Report
This paper presents a subspace deconvolution scheme for beamforming. It currently suffers from many drawbacks summarized as follows.
- The expansion of DOA is flipped as “Arrival of Direction” in many places.
- Many sentences need to be paraphrased to remove grammatical errors.
- Matrices are usually boldfaced but not italicized. The equations in this paper do not follow such mathematical conventions.
- Many figures and tables are divided into two pages, making them hard to read.
- The resolution of the figures must be enhanced.
- The computational complexities of the algorithms must be analyzed and compared.
- The proposed work needs to be compared with the latest counterparts other than the conventional ones.
- The optimality of the proposal needs to be proven rigorously.
Author Response
Dear Editor and Reviewers,
We submitted a manuscript in Feb.2022, and the paper was recommended a major revision. Based on the helpful comments, we have revised the paper.
In the main text of the revised version, the main revisions are highlighted with red bold. Please see the attachment.
We sincerely hope the revised manuscript meets the requirements of the reviewers.
Best regards!
Xiruo Su
College of Biomedical Engineering & Instrument Science,
Zhejiang University, Hangzhou, 310058, China.
E-mail: xiruo_su@zju.edu.cn

Round 2
Reviewer 2 Report
There exist several concerns that necessitate further improvements.
- While the notations of matrices have been corrected, those of vectors are still incorrect. Vectors are usually lowercased and boldfaced. In addition, the dimension of matrices and vectors need to be specified. The mathematical formulation should be carefully checked.
- In Table 2, the units of measures should be written differently. In the current form, for example, “Time/s” seems like the time divided by s.
- How did you measure the computation times? The environment needs to be clarified.
Author Response
Dear Reviewer,
Thanks for taking your time to review this manuscript. We really appreciate all your comments and suggestions! Please find my itemized responses in below and my revisions in the re-submitted files. In the main text of the revised version, the main revisions are highlighted with red bold and the modified formula format remains in black. Please see the attachment. Thanks again!
In the revised version, we have corrected some clarifications and analysis. The main revisions can be summarized as below:
- Corrected the notations of vectors and added the dimension of matrices and vectors.
- Corrected the expression of unites of measures in Table 2.
- Added a clarification of the environment to measure the computation times.
- Modified some clarification for the method description and simulation analysis.
- Re-described some English language styles.
Listed below are our responses to reviewers’ comments.
Reviewer#2 (1st Comment): While the notations of matrices have been corrected, those of vectors are still incorrect. Vectors are usually lowercased and boldfaced. In addition, the dimension of matrices and vectors need to be specified. The mathematical formulation should be carefully checked.
Response: Thank you for your helpful comments. It’s the first time for me to write an article and some formula formats have been ignored. In the revised version, I have carefully checked and corrected all formula errors. Your comments all mean a lot to me. Thanks again!
Reviewer#2 (2nd Comment): In Table 2, the units of measures should be written differently. In the current form, for example, “Time/s” seems like the time divided by s.
Response: We appreciate the detailed suggestion. We have changed all “/ Units” to “(Units)” in all Tables. Hopefully it will be easier for reading.
Reviewer#2 (3rd Comment): How did you measure the computation times? The environment needs to be clarified.
Response: Thank you for the good suggestion. We have added the environment to measure the computation times in the description of the experiment.
We sincerely hope the revised manuscript meets the requirements of publication.
Best regards!
Xiruo Su
College of Biomedical Engineering & Instrument Science,
Zhejiang University, Hangzhou, 310058, China.
E-mail: xiruo_su@zju.edu.cn

Round 3
Reviewer 2 Report
The manuscript looks better than before. Thank you.